# OpenReview forum: "SafeHarbor: Defining Precise Decision Boundaries via Hierarchical Memory-Augmented Guardrail for LLM Agent Safety"
_ICML.cc/2026/Conference — ICML 2026 regular_

### Official Review · Reviewer_hGG9 · 2026-03-02

**Soundness:** 3
**Presentation:** 2
**Significance:** 2
**Originality:** 3
**Overall Recommendation:** 4
**Confidence:** 4

**Summary:**

This paper proposes SafeHarbor, a safety framework designed to defend against harmful requests in agentic workflows. The primary goal of the framework is to mitigate misuse by malicious actors while preserving the usability and functionality of the agent.

SafeHarbor improves the generalization of defense rules by generating adversarial content enhanced through information-entropy-based clustering. It further incorporates an embedding-based memory system to manage and organize safety rules. Finally, the framework determines the safety of incoming queries by combining projection from contrastive learning with LLM-based judgment.

**Compliance With Llm Reviewing Policy:**

Affirmed.

**Final Justification:**

The rebuttal improves the paper. I maintain the positive score.

**Key Questions For Authors:**

1. Generalization to Unseen or Out-of-Distribution Attacks.
SafeHarbor constructs its defense framework using existing agent-scene datasets. How does the method generalize to unseen or distribution-shifted harmful queries that are not covered in the training data? In particular, do the authors have empirical evidence or theoretical justification supporting robustness beyond the observed attack patterns?

2. Fairness of Comparison with AgentAlign (SFT).
AgentAlign (SFT) appears to be trained on the original dataset rather than on data augmented via attack generation. Could the authors clarify whether this constitutes a fair comparison? If AgentAlign were trained on similarly enhanced or adversarially generated data and then used as a defense mechanism, would SafeHarbor performance remain superiority?

3. Training Efficiency Analysis.
The efficiency analysis currently focuses on inference efficiency. Could the authors also provide a comparison of training efficiency across methods, such as computational cost, training time, or resource requirements?

**Limitations:**

The authors do not discuss the limitations of their work. It is recommended that they provide a more thorough discussion of the limitations of the proposed framework.

**Strengths And Weaknesses:**

**Strengths:**
1. The paper presents a multi-module framework with clearly defined components that interact in a structured manner. The authors conduct comprehensive ablation studies to isolate and quantify the contribution of each module, which helps clarify the role of the proposed techniques and improves the transparency of the design.

2. The resulting defense system demonstrates coherent integration across modules and achieves reasonable resource efficiency. The framework appears to balance defensive capability with computational overhead, which is important for practical deployment in agentic environments.

**Weaknesses:**
1. The evaluation is conducted on a limited number of relatively dated models. Given the rapid progress in agent safety capabilities, particularly in systems released after GPT-4o, more recent models may already exhibit stronger robustness to harmful queries, which could lead to metric saturation. While this does not invalidate the reported results, comparisons with stronger and more contemporary models would make the empirical evidence more convincing. Additional experiments on advanced LLMs or modern agent systems, even at smaller scale, would further strengthen the reliability and relevance of the conclusions.

2. The baselines for agent defense are relatively simple. Although the paper discusses conceptual differences from ShieldAgent, it does not provide a direct empirical comparison. This is particularly important because both approaches appear to share similarities in their use of embeddings and memory mechanisms. A more explicit comparison with recent agent defense frameworks, especially those employing similar architectural ideas, would better highlight the distinct contributions and advantages of the proposed method.

3. In addition, more recent agent defense frameworks developed after ShieldAgent should be discussed to better position this work within the evolving literature. The paper would benefit from a clearer comparison with these approaches, including an analysis of their methodological differences and whether they are directly comparable to SafeHarbor in scope and objectives, such as AGrail [1]. Discussing how SafeHarbor differs from or improves upon such recent systems would help clarify its novelty and strengthen the justification of its core contributions.

[1] Luo, W., Dai, S., Liu, X., Banerjee, S., Sun, H., Chen, M., and Xiao, C. AGrail: A lifelong agent guardrail with effective and adaptive safety detection. In ACL 2025.

---

> ### Author Rebuttal · Authors · 2026-03-29
>
> We thank Reviewer #4 for appreciating our framework design, efficiency, and comprehensive evaluations, as well as the constructive feedback on baselines and fairness.
>
> # W1. Contemporary Models & Saturation
> Supplementary experiments on frontier models confirm SafeHarbor consistently optimizes the safety-utility trade-off across diverse architectures.
>
> | Base Model | Defense | Harmful Refusal (%) $\uparrow$ | Harmful Score (%) $\downarrow$ | Benign Refusal (%) $\downarrow$ | Benign Score (%) $\uparrow$ |
> | :--- | :--- | :---: | :---: | :---: | :---: |
> | **GPT-5** | Base | 69.3 | 16.8 | 2.8 | 69.4 |
> | | + SafeHarbor | **84.9 (+15.6)** | **8.7 (-8.1)** | 5.6 (+2.8) | 68.1 (-1.3) |
> | **Claude-3.6-Sonnet** | Base | 80.1 | 8.8 | 18.7 | 59.7 |
> | | + SafeHarbor | **84.1 (+4.0)** | **7.8 (-1.0)** | **14.8 (-3.9)** | **62.1 (+2.4)** |
> | **Qwen3-32B** | Base | 40.8 | 42.7 | 1.1 | 82.1 |
> | | + SafeHarbor | **94.3 (+53.5)** | **4.2 (-38.5)** | 17.6 (+16.5) | 65.7 (-16.4) |
>
> These results demonstrate that SafeHarbor effectively fortifies safety boundaries in open-source models (e.g., Qwen3-32B harmful refusal increases by $53.5\%$) while mitigating over-refusal in cautious models (e.g., Claude-3.6-Sonnet recovers $3.9\%$ benign utility).
>
> # W2. Positioning & Baseline Comparison
> Regarding AGrail, its optimization for OS-level Web Agents makes it incompatible with our trajectory benchmarks. However, because AGrail’s safety component relies on the same static rule logic as LLaMA-Guard, our evaluation against LLaMA-Guard  could cover the safety-rule-driven scenarios it addresses.
>
> For ShieldAgent, the lack of open-source code prevents a direct empirical comparison. ShieldAgent relies on a flat memory clustering structure and a single-layer Markov Logic Network for inference, which can lead to imprecise retrieval and rigid misclassifications.
> In contrast, SafeHarbor utilizes adversarial augmentation for broader semantic coverage and introduces a hierarchical memory architecture for precise, fine-grained retrieval.
>
> # Q1. Generalization to OOD Attacks
> We emphasize that our primary experimental design inherently constitutes a cross-dataset evaluation. SafeHarbor constructs its memory using AgentAlign, while testing is conducted on entirely different benchmarks.
>
> Furthermore, to explicitly address Out-of-Distribution (OOD) threats, SafeHarbor natively supports online memory updates for continuous rule evolution. To validate this mechanism, we simulated an online evolution scenario. Specifically, we reconstructed the memory tree by incrementally sampling varying quantities of harmful trajectories and their corresponding labels from AgentAlign. By continuously absorbing these patterns, the defense performance stabilizes at approximately 1,000 injected samples, achieving an optimal balance with a $3.0\%$ harmful score and $48.5\%$ benign score.
>
> | Injected Samples | Harmful Refusal (%) $\uparrow$ | Harmful Score (%) $\downarrow$ | Benign Refusal (%) $\downarrow$ | Benign Score (%) $\uparrow$ |
> | :--- | :---: | :---: | :---: | :---: |
> | 250 | 87.5 | 5.3 | 19.3 | 42.9 |
> | 500 | 86.4 | 4.4 | 15.9 | 45.8 |
> | 1000 | 88.6 | **3.0** | **8.0** | **48.5** |
> | 1500 | **89.2** | 3.3 | 14.2 | 44.2 |
> | 2000 | **89.2** | 3.7 | 14.2 | 42.7 |
>
> # Q2. Fairness of SFT Comparison
> We thank the reviewer for suggesting an SFT baseline evaluation with data augmentation. To ensure a fair comparison, we conducted a supplementary SFT experiment on Qwen-2.5 7B using the adversarially augmented dataset applied in SafeHarbor.
> | Method | Harmful Refusal (%) $\uparrow$ | Harmful Score (%) $\downarrow$ | Benign Refusal (%) $\downarrow$ | Benign Score (%) $\uparrow$ |
> | :--- | :---: | :---: | :---: | :---: |
> | AgentAlign (SFT) - Original | 90.3 | 1.4 | 11.9 | 20.7 |
> | AgentAlign (SFT) - Augmented | 58.5 | 7.8 | 2.3 | 23.0 |
> | + SafeHarbor | 89.2 | 3.9 | 9.1 | 49.4 |
>
> During evaluation, the augmented SFT model suffered from formatting corruption, frequently falling into infinite tool-calling loops. In contrast, SafeHarbor maintains a strictly superior safety-utility trade-off. We will include the corresponding analysis in the revised manuscript.
>
> # Q3. Training Efficiency Analysis
> Providing a strict direct comparison of training time is challenging due to fundamentally different paradigms. We detail the training/construction costs below:
>
> * Llama-Guard: Specific overheads are not open-sourced.
> * AgentAlign: SFT via LoRA on 18,749 samples took ~3.5 hours on two A100 GPUs.
> * A-Mem: Original trajectory self-evolution took ~10 hours for 7,000 trajectories, while our supplementary rule-only evolution took 75.5 minutes for 2,303 rules.
> * SafeHarbor: Building the hierarchical memory for all 4,956 original harmful samples took only ~45 minutes, as detailed below.
>
> | Processed Samples | Clusters | Incremental Time (min) | Cumulative Time (min) |
> | :--- | :---: | :---: | :---: |
> | 1000 | 357 | 4.48 | 9.39 |
> | 3000 | 727 | 9.33 | 26.51 |
> | 4956 | 1031 | 8.76 | 44.62 |

---

> > ### Author Rebuttal · Reviewer_hGG9 · 2026-04-01
> >
> > Thank you for the your response. Although I still have some concerns about the system’s real-world applications, such as adversarial attacks and tool-use scenarios, the other issues have been addressed. I will maintain the positive score.

---

### Official Review · Reviewer_kHbM · 2026-03-10

**Soundness:** 3
**Presentation:** 2
**Significance:** 3
**Originality:** 3
**Overall Recommendation:** 4
**Confidence:** 4

**Summary:**

This paper presents SafeHarbor, an approach that learns precise decision boundaries and supports fast online inference to judge whether a user input is safe. Specifically, it first generates safety rules from seed trajectories, forming attack memory trees, and designs benign exemptions from benign trajectories. It then trains a safety projector to quickly and coarsely distinguish harmful queries from benign ones. Finally, during online inference, it either follows a fast path based on safety and harmfulness scores or uses the stored rules to instruct an LLM to make the judgment. Across the empirical evaluation, it performs reasonably well on the trade-off between helpfulness and harmlessness.

**Compliance With Llm Reviewing Policy:**

Affirmed.

**Final Justification:**

Most of my concerns are addressed. I will maintain my positive score

**Key Questions For Authors:**

* Could you extend the baseline comparison to include gpt-oss-20b-safeguard, which may be faster and more flexible, and may reduce over-refusal with carefully constructed prompts?
* How can the current approach be extended to more complex real-world scenarios, especially code agents or computer-use agents? What are the main challenges in that setting?
* How is A-Mem implemented in this scenario? If it were rewritten to explicitly ask for safety rule capture rather than using its original memory update rule, would it also perform strongly?

* On the right side of line 233 and line 241, what is the $\beta$ dataset? If it refers to benign data, why is the loss computed only on this dataset?

**Limitations:**

See weakness above

**Strengths And Weaknesses:**

# Strength
* This paper presents an efficient guardrail for LLM agents.
* This paper conducts a comprehensive evaluation showing that it balances both utility and safety.
* The ablation studies reveal interesting insights into the importance of each component in the overall pipeline.
# Weakness
* The current testing scenario is too simple. In real, more complex agents, such as code agents (e.g., Claude Code) and computer-use agents (e.g.,, OpenClaw), the situation is often much more complex, and execution safety depends heavily on the actual trajectory and environment, which can be extremely long, rather than on the user query alone or on synthetic short trajectories. In this case, the current approach seems limited.
* The models evaluated in this paper do not reflect the current state of the art in agentic capability. Further experiments on models such as the GPT-5 series or Claude 4.6 would make the evaluation stronger.

---

> ### Author Rebuttal · Authors · 2026-03-29
>
> We sincerely thank Reviewer #3 for the positive evaluation of our efficiency, comprehensive evaluation, and ablation insights. We appreciate the constructive feedback regarding the extension to complex agentic scenarios and the precision of our technical presentation.
>
> # Scenario Complexity & Extension (Addressing Weakness 1 & Q2)
> We appreciate the reviewer’s perspective on scenario complexity. First, we wish to clarify that our benchmarks, particularly AgentHarm, are far from simple. As demonstrated in Appendix Figure 9, queries often necessitate multi-turn interactions where the LLM must accurately navigate dozens of available tools, confirming that we address real-world logical consistency challenges.
>
> Regarding the extension to computer-use agents (e.g., Claude Code), the primary challenge lies in the state-space explosion. SafeHarbor is designed as an atomic-level guardrail; by ensuring each individual action adheres to the "Prohibition/Exemption" logic, it provides a robust foundation for multi-turn safety. To handle greater complexity, our hierarchical memory can naturally extend to "state-conditioned rules" (e.g., a code-deletion rule could exempt "temporary build paths"). We view SafeHarbor as a critical first-line defense architecturally ready to integrate with environment-monitoring systems.
>
> # Frontier Models & Baselines (Addressing Weakness 2 & Q1)
> We thank the reviewer for the constructive suggestion to evaluate SafeHarbor on frontier models. Supplementary experiments (Table R1) show that our framework consistently optimizes the safety-utility trade-off across heterogeneous architectures.
>
> | Base Model | Defense | Harmful Refusal (%) $\uparrow$ | Harmful Score (%) $\downarrow$ | Benign Refusal (%) $\downarrow$ | Benign Score (%) $\uparrow$ |
> | :--- | :--- | :---: | :---: | :---: | :---: |
> | **GPT-5** | Base | 69.3 | 16.8 | 2.8 | 69.4 |
> | | + SafeHarbor | **84.9 (+15.6)** | **8.7 (-8.1)** | 5.6 (+2.8) | 68.1 (-1.3) |
> | **Claude-3.6-Sonnet** | Base | 80.1 | 8.8 | 18.7 | 59.7 |
> | | + SafeHarbor | **84.1 (+4.0)** | **7.8 (-1.0)** | **14.8 (-3.9)** | **62.1 (+2.4)** |
> | **Qwen3-32B** | Base | 40.8 | 42.7 | 1.1 | 82.1 |
> | | + SafeHarbor | **94.3 (+53.5)** | **4.2 (-38.5)** | 17.6 (+16.5) | 65.7 (-16.4) |
>
> These results demonstrate that SafeHarbor effectively fortifies open-source models (e.g., Qwen3-32B harmful refusal increases by 53.5%) while simultaneously mitigating over-refusal in inherently cautious models (e.g., Claude-3.6-Sonnet recovers 3.9% benign utility). Regarding the suggested comparison with gpt-oss-safeguard-20b, we clarify it is a specialized moderation model lacking the native tool-calling capabilities required for agentic benchmarks, making direct comparison technically unsuitable. We position SafeHarbor as a complementary layer enhancing such specialized safety infrastructures.
>
> # A-Mem Implementation & Logic (Addressing Q3)
> We thank the reviewer for this insightful question. In our original setup, A-Mem maintained full historical trajectories. Due to memory locking during updates, this process is computationally intensive, taking ~10 hours for 7,000 samples. However, as detailed in Table 1 of the main text, this original trajectory-maintenance approach is still highly effective for models with strong reasoning capabilities like GPT-4o, achieving a high harmful refusal rate of 86.9% and a benign score of 61.3, while attaining the optimal benign refusal rate. To directly address your query, we rewrote A-Mem to explicitly capture and manage safety rules using its native evolution mechanism.
>
> | Method | Harmful Refusal (%) $\uparrow$ | Harmful Score (%) $\downarrow$ | Benign Refusal (%) $\downarrow$ | Benign Score (%) $\uparrow$ |
> | :--- | :---: | :---: | :---: | :---: |
> | Rule-adapted A-Mem | 51.1 | 24.4 | **0.6** | 49.2 |
> | SafeHarbor | **89.2** | **3.9** | 9.1 | **49.4** |
>
> While the rule-adapted A-Mem maintains high benign utility, its defensive capability (51.1%) falls significantly short of SafeHarbor (89.2%). This demonstrates that relying purely on A-Mem's flattened evolution mechanism is insufficient for establishing robust boundaries. In contrast, SafeHarbor's two-tier hierarchical tree processes all 4,956 harmful samples in just 22.5 minutes via cluster-level multithreading, confirming our structural and operational advantages.
>
> # Technical Notation & Formulation (Addressing Q4)
> We thank the reviewer for their meticulous reading and have corrected Eq. 12 and Eq. 13 for technical precision:
> 1. We replaced $x$ in Eq.12 and Eq. 13 with $z$ to consistently represent latent feature embeddings.
> 2. We clarified that $\mathcal{B}$ refers to a mixed mini-batch of benign and harmful samples.
> 3. The loss is computed across the entire mixed batch to ensure a balanced decision boundary. This phrasing oversight from an earlier draft has been updated in Section 3.5 to ensure full internal consistency.

---

> > ### Author Rebuttal · Reviewer_kHbM · 2026-03-31
> >
> > Thank you for your response. While I still have some concerns about applying this as a real-world monitoring system, my other questions have largely been resolved. I will maintain my positive score.

---

### Official Review · Reviewer_uru6 · 2026-03-12

**Soundness:** 3
**Presentation:** 2
**Significance:** 3
**Originality:** 3
**Overall Recommendation:** 4
**Confidence:** 4

**Summary:**

This paper studies safety for tool-using LLM agents, with the goal of reducing over-refusal while still blocking harmful agentic actions. The proposed method, SAFEHARBOR, combines three components: adversarial rule generation from harmful trajectories, a hierarchical memory that stores paired harmful prohibitions and benign exemptions, and an online inference pipeline that uses a lightweight safety projector plus retrieval for fast filtering and falls back to an LLM judgment stage for ambiguous cases (Sec. 3). The paper evaluates the method on AgentHarm and AgentSafetyBench across GPT-4o, Mistral-8B, and Qwen-2.5-7B, and reports comparisons to prompting, memory-based, classifier-based, and SFT baselines, together with ablations and efficiency analysis (Sec. 4-5).

**Compliance With Llm Reviewing Policy:**

Affirmed.

**Final Justification:**

The rebuttal addresses my concern; I remain with a weak accept; I assume the author will handle the manuscript appropriately in their revision.

**Key Questions For Authors:**

Section 4.1 states that the memory-construction dataset has 18,749 instances, including 9,783 benign and 4,956 harmful samples. What accounts for the remaining instances? Clarifying this would improve confidence in the experimental setup.

The paper presents SAFEHARBOR as training-free or plug-and-play, yet the method trains a safety projector. Can the authors clarify whether the intended claim is no base-model fine-tuning rather than fully training-free? A precise answer would improve the technical framing of the paper.

How sensitive are the main results to the domain-specific LLM judgment prompt in Appendix H? In particular, do the gains persist with a domain-neutral prompt, or in non-administrative agent settings? A strong answer here would improve my view of the method’s generality.

Please clarify the relation to the recent A-MemGuard line of work. My current reading is that A-MemGuard focuses on memory poisoning and self-correcting lesson memory, whereas SAFEHARBOR focuses on action-time safety boundary definition for current tool-use requests. If that is the intended distinction, I encourage the authors to state it explicitly and explain whether the two methods are complementary rather than directly competing.

Please provide more detail on the latency/VRAM protocol and, if available, repeated-run variability. This would make the efficiency claims easier to interpret and strengthen the empirical soundness.

**Limitations:**

No. The paper includes an impact statement, but it is too dismissive. I would encourage the authors to discuss more directly: the dual-use risk of releasing attack-enhancement templates, the domain specificity of the LLM judgment prompt, the dependence on tuned thresholds, and the extent to which the method’s strongest results on smaller backbones rely on a stronger verifier.

**Strengths And Weaknesses:**

The paper addresses an important practical problem: agent safety systems should not only block harmful actions, but also avoid rejecting legitimate high-risk-looking administrative or multi-step tool-use requests. The proposed system is coherent at the design level. In particular, SAFEHARBOR does not simply retrieve similar examples; it stores paired prohibition/exemption rules, uses a hierarchical memory for scalable retrieval, and combines a learned safety projector with an LLM judgment stage. The empirical evaluation is reasonably broad, covering two benchmarks, several backbone models, multiple baseline families, and component ablations. Overall, the results support the claim that the method often achieves a better safety-utility trade-off than the compared baselines (Sec. 3-5).

My main concern is technical framing. The paper repeatedly emphasizes training-free or plug-and-play deployment, but the method also trains a safety projector with a BCE plus contrastive objective and tuned hyperparameters (Sec. 3.5, App. B-C). I therefore think the more precise claim is no base-model fine-tuning rather than fully training-free. A second concern is a concrete data-accounting inconsistency: Sec. 4.1 states that the memory-construction dataset contains 18,749 instances, including 9,783 benign and 4,956 harmful samples, but these numbers do not sum to the stated total. This should be clarified. A third concern is that the LLM judgment prompt in the appendix is strongly domain-specific: it assumes an Authorized Administrator and a Senior Technical Auditor, and explicitly adopts a Presumption of Utility.” This makes the method look more tailored to administrative-security scenarios than the main paper currently acknowledges. Finally, the main tables report point estimates only; I did not find variability statistics, and Table 4 would be easier to interpret with clearer measurement details such as hardware and batching assumptions.

Presentation is generally solid but not fully polished. The method narrative is readable, Figure 2 is helpful, and the appendix substantially improves reproducibility by exposing prompts and implementation details. However, there are avoidable inconsistencies and wording issues, including the table typo Ministral 8B-Instruct, the overloaded description of 𝜆, and several awkward phrases. These do not prevent understanding, but they weaken the precision of the presentation.

On originality, I view the contribution as a thoughtful systems combination rather than a new paradigm. The pairing of harmful prohibitions with benign exemptions inside a hierarchical memory is the most distinctive design choice, and the ablations suggest that this combination matters. At the same time, the authors should position the work more clearly relative to recent memory-security defenses, such as A-MemGuard [1]. By contrast, SAFEHARBOR focuses on action-time decision boundaries for current tool-using requests, using synthesized rule/exemption retrieval, a safety projector, and an LLM judge, and evaluates on AgentHarm and AgentSafetyBench. These are adjacent but not redundant contributions, and the paper would be stronger if this distinction were made explicit.



[1] A-MemGuard: A Proactive Defense Framework for LLM-based Agent Memory (https://arxiv.org/abs/2510.02373)

---

> ### Author Rebuttal · Authors · 2026-03-29
>
> We thank Reviewer #2 for the constructive feedback. We have addressed all technical and framing inconsistencies in the revised manuscript to improve overall precision.
> # Q1: Data Accounting Inconsistency
> We apologize for the lack of clarity regarding our dataset in Section 4.1. Per the AgentAlign distribution, the 18,749 instances consist of 9,783 benign, 4,956 harmful, and 4,010 neutral trajectories. We categorized these neutral samples as benign. We will explicitly include this detailed breakdown in the revised manuscript.
> # Q2: Technical Framing: "Training-free" vs. "No Base-model Fine-tuning"
> We appreciate the reviewer’s precision and agree that "no base-model fine-tuning" more accurately describes our framework. The Safety Projector is a highly efficient two-layer MLP ($384 \rightarrow 512 \rightarrow 128$) operating on top of a frozen, pre-trained Sentence Transformer. This architecture restricts optimization to approximately 0.26 million parameters, requiring zero gradient updates to the billion-parameter base LLM. Empirically, training on 5,948 triplets for 15 epochs completes in just ~11.3 seconds on a single NVIDIA A100 (80GB) GPU, with the inference latency added by the MLP being statistically negligible. This rapid optimization confirms that SafeHarbor retains "plug-and-play" advantages while providing significantly sharper decision boundaries than pure retrieval-based approaches.
>
> # Q3: Domain Specificity of the LLM Judgment Prompt
> We thank the reviewer for the insightful comments on our prompt framing. We clarify the fundamental distinction between the Adversarial Generation stage (Section 3.3) and the LLM Judgment module. As detailed in Section 3.3, our attack enhancement adopts an LLM-driven approach where the model functions as a strategic attacker, dynamically selecting the optimal paradigm based on the context to maximize attack success. Consequently, Figure 7 represents the adversarial template for privilege-escalation attacks and does not constitute the final judgment logic.
>
> For evaluation, the "Auditor-Administrator" relationship in Figure 13 is a functional abstraction designed to simulate a professional auditing environment. This allows the LLM Judge to evaluate tool-use requests against retrieved hierarchical rules while enforcing the "Presumption of Utility" without granting blind trust. Our consistent success across 11 diverse AgentHarm categories—including Fraud, Cybercrime, and Harassment—demonstrates that this framing is domain-agnostic and effectively generalizes across varied agentic environments.
>
> # Q4: Relation to A-MemGuard and A-Mem Adaptation
> We appreciate the reference to A-MemGuard (Wei et al., 2025), but wish to clarify a potential misunderstanding regarding our baseline. The A-Mem evaluated in our experiments is the general-purpose memory framework proposed by Xu et al. (NIPS 2025). While A-MemGuard prioritizes defending against memory poisoning to ensure record integrity, SafeHarbor specifically addresses the over-refusal problem in real-time tool-use scenarios by precise Prohibition-Exemption boundaries. These two works target complementary stages of the agent security lifecycle.
>
> To evaluate the defensive potential of the general-purpose A-Mem framework, our original baseline sampled 8,000 instances from AgentAlign. We also have conducted a supplementary experiment. We rewrote A-Mem to explicitly capture and manage safety rules using its native flat evolution mechanism.
>
> | Method | Harmful Refusal (%) $\uparrow$ | Harmful Score (%) $\downarrow$ | Benign Refusal (%) $\downarrow$ | Benign Score (%) $\uparrow$ |
> | :--- | :---: | :---: | :---: | :---: |
> | Rule-adapted A-Mem | 51.1 | 24.4 | **0.6** | 49.2 |
> | SafeHarbor | **89.2** | **3.9** | 9.1 | **49.4** |
>
> The results indicate that while the rule-adapted A-Mem maintains excellent benign utility, its defensive capability (51.1% harmful refusal) falls significantly short of SafeHarbor (89.2%).
>
> # Q5: Efficiency Protocol and Variability
> We thank the reviewer for the request to clarify our efficiency evaluation protocol. All experiments were conducted on an A100 (80GB), with open-source models deployed via vLLM and proprietary models accessed via APIs. Table 4 reports the mean of five independent runs using Qwen-2.5-7B as the agent backbone. Unlike LlamaGuard, which requires an external 8B-parameter classification model (doubling VRAM to 30GB), SAFEHARBOR’s overhead is restricted to a 0.26M-parameter MLP and a fast vector-retrieval step. This design maintains a minimal 14GB footprint and 306.67ms latency. We will include these in the revised Implementation Details.
>
> # Q6: Presentation and Typos
> We thank the reviewer for the meticulous proofreading and for identifying these inconsistencies. We have corrected the nomenclature to Mistral-8B-Instruct and disambiguated the notation for $\lambda$  to ensure mathematical precision. A comprehensive proofreading pass has been completed in the revised manuscript.

---

> > ### Author Rebuttal · Reviewer_uru6 · 2026-04-03
> >
> > Thank you for the detailed rebuttal. The data accounting issue is now clear, the training-free framing is corrected, and the supplementary A-Mem experiment is helpful. The efficiency protocol details are also appreciated. My positive score remains unchanged.
> >
> > In addition to the content that the authors promise to do so, one more suggestion for the revision is: the related work section would benefit from a more explicit discussion of how SafeHarbor relates to adjacent memory defense work. That said, what overlaps, what appears related but is actually distinct, and where the boundaries lie. This would help readers more clearly situate the contribution.

---

### Official Review · Reviewer_j541 · 2026-03-15

**Soundness:** 2
**Presentation:** 3
**Significance:** 2
**Originality:** 2
**Overall Recommendation:** 4
**Confidence:** 3

**Summary:**

The paper proposes SafeHarbor, a novel safety framework for agents designed to solve the "over-refusal" problem—where strict safety guardrails inadvertently block legitimate, benign tasks. Unlike traditional static guardrails that use rigid decision boundaries, SafeHarbor utilizes a hierarchical memory-augmented system that extracts context-aware defense rules through adversarial generation. This allows the system to dynamically distinguish between subtle malicious attacks and complex but safe user requests in real-time. Extensive testing on models like GPT-4o shows that the framework maintains high safety standards, achieving a harmful refusal rate of over 93%, while significantly improving utility by preserving a benign task success rate of 63.6% with minimal computational latency.

**Compliance With Llm Reviewing Policy:**

Affirmed.

**Final Justification:**

Given additional clarifications and evaluations, I think the paper can be a convincing publication with minor revision.

**Key Questions For Authors:**

Please refer to the weakness part.

**Limitations:**

Yes, the limitation is discussed

**Strengths And Weaknesses:**

Strengths of the Submission
- Highly Relevant and Timely Problem: The paper addresses a critical bottleneck in the real-world deployment of LLM agents: the "over-refusal" problem. Balancing rigorous safety against malicious trajectories with the utility needed to complete complex, benign tasks is a highly relevant challenge for the ICML community.
- Novel and well-structured Framework: The proposed SAFEHARBOR architecture is elegant and comprehensive. The three-stage pipeline—combining offline adversarial rule generation, a hierarchical memory tree with information-gain-based updates, and an efficient online dual-path routing mechanism—offers a strong systemic approach to agent safety.

Weakness:
1. Insufficient Analysis of the Gating Mechanism (Latent vs. LLM Judge)

While the framework innovatively combines a lightweight latent-based safety projector with an LLM judge, the exact dynamics of this integration require deeper exploration. Specifically, under what exact semantic or contextual conditions does the latent pre-filter fail, necessitating the LLM judge? A more granular analysis of the failure cases for each routing path—and a clear discussion on when a feature-based judge is preferable to an LLM judge—would significantly consolidate the paper's theoretical grounding.

2. Ambiguous Terminology Regarding the "Memory Tree"

The terminology "Memory Tree" slightly overstates the structural complexity of the module. Based on the methodology, it appears to function more as a two-level hierarchical cluster (where broad risk categories serve as internal routing nodes, and fine-grained attacks/rules serve as leaf nodes) rather than a deep, multi-level tree structure. Clarifying this naming convention or better justifying the "tree" designation would prevent reader confusion.

3. Clarification of "LLM Self-Evolution" vs. Guardrail Update

The concept of Info-Gain-based memory updating is a clear and novel application for adversarial evolution. However, the phrasing "LLM self-evolution" is misleading. Since the guardrail trajectories and rules should ideally be agnostic to the specific base LLM of the agent, the "evolution" applies strictly to the guardrail's external memory database, not the agent's internal policy or weights. The authors should clarify this distinction to avoid implying that the base model itself is undergoing continuous alignment.

4. Baseline Model Recency

Given the incredibly rapid pace of foundation model development and the ICML 2026 timeline, evaluating on slightly older backbones limits the immediate impact of the empirical claims. To ensure the paper reflects the current state-of-the-art, the authors should ideally include evaluations on the most recent flagship models (e.g., the GPT-5 or Qwen-3 series), assuming they were accessible prior to the submission deadline.

5. Lack of Empirical Validation for "Online" Adaptation

The methodology heavily emphasizes the "online stage" and the continuous, dynamic evolution of the memory structure. However, the experimental section evaluates the framework statically. To substantiate the claims regarding self-organizing continuous optimization, the authors need to include longitudinal evaluations demonstrating how the framework's performance (both utility and safety) improves over time as the memory evolves against a continuous stream of online queries, compared to static baseline methods.

---

> ### Author Rebuttal · Authors · 2026-03-29
>
> We thank Reviewer #1 for the constructive feedback and for recognizing the elegance and systemic importance of our SAFEHARBOR architecture in addressing over-refusal. We have carefully addressed all noted weaknesses in the revised manuscript.
>
> # Q1: Analysis of the Gating Mechanism
>
> We appreciate the push for clarity on our routing logic. Our selection of thresholds ($\tau_{harmful} = 0.2, \tau_{benign} = 0.65$) is dictated by the zero-tolerance safety principle visualized in Figure 5 of Appendix F. As shown in Figure 5(a), these conservative settings effectively compress harmful leakage to a statistically negligible level (near zero). While the latent projector is prioritized for its efficiency—successfully offloading 23% to 25% of traffic via the FastPath—it reaches its reasoning limit on queries where malicious intent is contextually obscured by seemingly legitimate tool invocations. In these instances, the projector provides a calibrated auxiliary signal, but the system routes the query to the LLM judge for deep logical scrutiny to resolve the ambiguity. This tiered approach ensures we maximize throughput for clear-cut benign scenarios while maintaining the strict decision boundaries established in our sensitivity analysis.
>
> # Q2: Terminology Regarding the "Memory Tree"
>
> We appreciate the reviewer pointing out this terminology nuance. "Memory Tree" could indeed imply a deep, multi-layered architecture rather than our targeted two-level approach (mapping semantic intents to specific Prohibition/Exemption rules). To accurately reflect this design and maintain consistency with our title, we will replace "Memory Tree" with "Hierarchical Memory Structure" throughout the revised manuscript.
>
> # Q3: Clarification of "Self-Evolution" vs. Guardrail Update
>
> We appreciate the reviewer’s feedback on our terminology. To ensure absolute precision, we will replace the broad term "LLM self-evolution" with "LLM Memory Self-Evolution" throughout the revised manuscript. As established by the taxonomy of Shao et al. (Misevolve, NIPS 2025), memory evolution is a fundamental sub-paradigm of LLM evolution. SafeHarbor specifically implements this by dynamically optimizing external hierarchical safety rules in-context while the base model parameters remain completely frozen.
>
> # Q4: Baseline Model Recency
>
> We thank the reviewer for the insightful suggestion to evaluate our framework against the latest frontier models. To demonstrate robust generalization beyond our initial baselines, we have conducted supplementary AgentHarm evaluations comparing the native and SafeHarbor-augmented versions of GPT-5, Claude-3.6-Sonnet, and Qwen3-32B. The empirical results confirm our framework's adaptability: it effectively mitigates the severe security vulnerabilities in the open-source model, increasing the harmful refusal rate by 53.5%. Furthermore, SafeHarbor successfully alleviates both the under-defensiveness in proprietary models by fortifying their safety boundaries and the over-refusal issues by correcting false positives in inherently cautious architectures. Crucially, our dynamic exemption rules ensure these safety gains are achieved while maintaining or even improving overall benign utility.
>
> | Base Model | Defense | Harmful Refusal (%) $\uparrow$ | Harmful Score (%) $\downarrow$ | Benign Refusal (%) $\downarrow$ | Benign Score (%) $\uparrow$ |
> |:---|:---|:-:|:-:|:-:|:-:|
> | GPT-5 | Base | 69.3 | 16.8 | 2.8 | 69.4 |
> | | + SafeHarbor | **84.9 (+15.6)** | **8.7 (-8.1)** | 5.6 (+2.8) | 68.1 (-1.3) |
> | Claude-3.6-Sonnet | Base | 80.1 | 8.8 | 18.7 | 59.7 |
> | | + SafeHarbor | **84.1 (+4.0)** | **7.8 (-1.0)** | **14.8 (-3.9)** | **62.1 (+2.4)** |
> | Qwen3-32B | Base | 40.8 | 42.7 | 1.1 | 82.1 |
> | | + SafeHarbor | **94.3 (+53.5)** | **4.2 (-38.5)** | 17.6 (+16.5) | 65.7 (-16.4) |
>
> # Q5. Empirical Validation for "Online" Adaptation
> We thank the reviewer for the insightful suggestion to evaluate the system’s dynamic evolution. To investigate this, we have conducted a longitudinal experiment on the AgentHarm benchmark using Qwen2.5-7B, progressively injecting raw attack samples from AgentAlign while ablating the Safety Projector and Attack Enhancement modules to isolate the memory scaling effect. The empirical results confirm that SafeHarbor reaches an optimal performance peak at 1,000 samples, beyond which injecting excessive raw attacks induces over-conservatism and increases benign refusals. We will incorporate this granular analysis into the revised Appendix to substantiate the efficacy of our memory evolution mechanism.
> | Injected Samples | Harmful Refusal (%) ↑ | Harmful Score (%) ↓ | Benign Refusal (%) ↓ | Benign Score (%) ↑ |
> | :--- | :---: | :---: | :---: | :---: |
> | 250 | 87.5 | 5.3 | 19.3 | 42.9 |
> | 500 | 86.4 | 4.4 | 15.9 | 45.8 |
> | 1000 | 88.6 | **3.0** | **8.0** | **48.5** |
> | 1500 | **89.2** | 3.3 | 14.2 | 44.2 |
> | 2000 | **89.2** | 3.7 | 14.2 | 42.7 |

---

> > ### Author Rebuttal · Reviewer_j541 · 2026-04-05
> >
> > I thank the authors for the detailed rebuttal. The additional evaluations and technical clarifications have significantly strengthened the manuscript. My remaining minor concern regarding the term 'memory self-evolution' is that since memory is inherently dynamic and subject to updates, this branding may be overemphasized. In my view, the more critical contribution is the concept of 'memory for safety,' a dimension that remains relatively underexplored in existing agent harness.

---

### Decision · Program_Chairs · 2026-04-30

**Decision:**

Accept (regular)

**Comment:**

This paper proposes SafeHarbor, a memory-augmented inference-time guardrail framework for LLM agent safety that addresses the over-refusal problem by combining adversarially generated context-aware rules with a hierarchical memory structure and a dual-path online routing mechanism. Prior to the rebuttal, reviewers broadly recognized the framework's practical relevance, coherent multi-module design, and comprehensive ablation studies, but raised shared concerns about the limited evaluation on relatively dated backbone models, the misleading "training-free" framing given the safety projector training, ambiguous terminology around "self-evolution," the domain-specificity of the LLM judgment prompt, and insufficient baselines against more recent agent defense frameworks. After the rebuttal and discussion, all four reviewers maintained their positive scores, with their major concerns largely resolved through supplementary experiments on frontier models, clarified technical framing, additional comparisons against rule-adapted A-Mem, and longitudinal memory evolution analysis; Reviewer kHbM and hGG9 noted residual concerns about real-world applicability to complex agentic environments, while Reviewer uru6 encouraged a more explicit related-work discussion on adjacent memory defense systems in the camera-ready version. By trading off the strengths and weaknesses of this paper, as well as the reviewers' unanimously positive post-rebuttal assessments, I decide to accept this paper.